# Advances in Understanding of the Role of Lipid Metabolism in Aging

**DOI:** 10.3390/cells10040880

**Published:** 2021-04-13

**Authors:** Ki Wung Chung

**Affiliations:** College of Pharmacy, Pusan National University, Busan 46214, Korea; kieungc@pusan.ac.kr; Tel.: +82-51-510-2819; Fax: +82-51-510-2821

**Keywords:** aging, lipid metabolism, age-related diseases

## Abstract

During aging, body adiposity increases with changes in the metabolism of lipids and their metabolite levels. Considering lipid metabolism, excess adiposity with increased lipotoxicity leads to various age-related diseases, including cardiovascular disease, cancer, arthritis, type 2 diabetes, and Alzheimer’s disease. However, the multifaceted nature and complexities of lipid metabolism make it difficult to delineate its exact mechanism and role during aging. With advances in genetic engineering techniques, recent studies have demonstrated that changes in lipid metabolism are associated with aging and age-related diseases. Lipid accumulation and impaired fatty acid utilization in organs are associated with pathophysiological phenotypes of aging. Changes in adipokine levels contribute to aging by modulating changes in systemic metabolism and inflammation. Advances in lipidomic techniques have identified changes in lipid profiles that are associated with aging. Although it remains unclear how lipid metabolism is regulated during aging, or how lipid metabolites impact aging, evidence suggests a dynamic role for lipid metabolism and its metabolites as active participants of signaling pathways and regulators of gene expression. This review describes recent advances in our understanding of lipid metabolism in aging, including established findings and recent approaches.

## 1. Introduction

Aging is a complex biological process characterized by accumulation of changes over time with loss of physiological integrity [1]. These changes are often associated with increased vulnerability to specific diseases, including those that are age-related. Although great advances brought about by research have widened our understanding of aging and age-related diseases, the underlying causes of aging remain poorly defined because of its complexity and multifarious nature. Lopez-Otin et al. suggested that nine candidate hallmarks contribute to the aging process [2]. These include major changes that occur during aging, such as stem cell exhaustion, genomic instability, and altered cellular communication. It was noted that these candidate hallmarks are not of equal importance in accelerating aging, because some changes occur to compensate for the aging process. The nine hallmarks include genomic instability, telomere attrition, epigenetic alterations, loss of proteostasis, deregulated nutrient sensing, mitochondrial dysfunction, cellular senescence, stem cell exhaustion, and altered intercellular communication. Furthermore, the hallmarks are interconnected, which makes it difficult to determine their exact role in aging. Nevertheless, advances in genetic technologies, pharmacological approaches, and dietary interventions have made it feasible to distinguish important targets of complex aging processes [2]. The identification and verification of target genes (or proteins) have helped us to understand the mechanisms underlying the aging process.

One notable cause of aging is changes in nutrient-sensing signaling pathways [3]. Throughout the aging process, deregulation of nutrient-sensing signaling pathways has been reported. Within nutrient-sensing, insulin-like growth factor-1 (IGF-1) and insulin signaling pathways are the most conserved aging-associated pathways [4]. Genetic approaches from shorter-lived yeast, worms, flies, and rodents have shown that the IGF-1 and insulin signaling pathways play pivotal roles in extending longevity [5]. The roles of downstream effectors in aging have also been validated in various species. Strikingly, AKT-mediated downstream signaling pathways leading to the Forkhead box O (FOXO) family of transcription factors have been linked to longevity in both humans and model organisms [6]. Interestingly, the anti-aging effect of calorie restriction (CR) is mediated by changes in the nutrient-sensing signaling pathway [7]. CR-mediated life extension involves the AKT–FOXO signaling pathway in various model organisms [8]. Furthermore, other nutrient-sensing signaling pathways, including mammalian target of rapamycin (mTOR), AMP-activated protein kinase (AMPK), and sirtuins, are also associated with aging and age-related diseases, and provide interesting targets for anti-aging research [9,10,11]. Overall, the anabolic signaling pathway is associated with accelerated aging, while decreased nutrient signaling pathways extend lifespan. This suggests that nutrient-sensing signaling exerts a profound effect on the aging process.

Changes in lipid metabolism play an important role in various pathophysiological conditions [12]. In the body, lipids are used as an immediate energy source or for later use [13]. Fatty acid oxidation (FAO), which mostly occurs in the mitochondria, converts fatty acids to acetyl-CoA. Metabolically active organs rely on this pathway, because FAO yields relatively high energy compared to other metabolites [14]. When lipid intake exceeds the amount required by the body, the excess will be stored in the adipose tissues in forms of triglycerides. Under normal conditions, adipose tissue deposits most of the surplus lipids in the body. However, under specific conditions, such as metabolic diseases and aging, increased lipotoxicity can lead to various age-related diseases, including cardiovascular disease, cancer, arthritis, type 2 diabetes, and Alzheimer’s disease [15,16,17,18,19]. Excessive lipids also can be stored in other tissues including the liver, muscle, and kidney [20]. This ectopic accumulation of lipids can occur due to excessive lipids in the body, and by altered metabolic signaling in the affected organs. Under metabolically disordered conditions, lipid storage and lipid profiles are altered. These changes were once considered to be consequences of metabolic changes, as passive components of systemic metabolism.

Studies have revealed that changes in lipid metabolism actively participate in various cellular processes [12,21,22,23]. Altered lipid metabolism may involve the following three mechanisms: (1) Deficient energy utilization due to altered lipid metabolic processes can induce cellular changes. Since efficient energy supply is critical for cellular function and survival, insufficient energy supply will have broad effects on cellular processes. Organs and tissues, depending on fatty acid oxidation, will have a greater effect. (2) Adipocytes in adipose tissue release several adipokines. The extent of adipokine release is dependent on the amount of lipids in the cells. For example, the amount of leptin hormones released is higher under conditions of high adiposity. Leptin suppresses appetite by activating its receptor in the hypothalamic region of the brain. However, because other cell types also express leptin receptors and respond to leptin, leptin signaling has been implicated in various pathophysiological processes. (3) Fatty acids and other lipid metabolites participate directly in cellular processes. Excessive fatty acids increase cellular stress inducing cytotoxicity. Furthermore, specific fatty acid metabolites, including eicosanoids and sphingolipids, also participate in cellular responses and are associated with age-related diseases.

The aim of this review is to summarize advances in our understanding of the role of altered lipid metabolism in aging. Because recent reviews have covered how lipid metabolites actively participate in the cellular process and their role in diseases, this review will focus on recent experimental evidence linking altered lipid metabolism to aging [24]. The main topics will include overall changes in lipid metabolism, changes in leptin signaling, and changes in lipid metabolites during aging. Furthermore, how altered lipid metabolism can affect epigenetics will also be discussed, as this also plays an important role in the aging process. Finally, the effects of anti-aging CR, or other interventions that affect lipid metabolism, will be discussed.

## 2. Evidence Linking Lipid Metabolism to Aging

Systemic changes in lipid metabolism have been well studied [24,25]. Overall lipid metabolism changes during aging, including the content of lipids in the organs and their transport between major organs. Aging increases the levels of plasma triglycerides along with increased plasma lipoproteins, while the rate of plasma triglyceride clearance decreases with decreased activity of lipoprotein lipase [26,27]. Recent studies have demonstrated the mechanisms through which lipolysis changes during aging. An age-induced reduction in lipolysis is associated with decreased availability of catecholamines in the adipose tissue [28,29]. The activity of hormone-sensitive lipase is reduced in aged adipose tissue which is usually stimulated by β-adrenergic receptor on adipocytes. Overall, these changes explain the changes in lipid metabolism and increases in adiposity in adipose tissue and plasma. However, although these can explain how changes in lipid metabolism occur at the systemic level, they do not fully explain how changes in lipid metabolism contribute to aging and age-related diseases.

In addition to systemic changes in lipid metabolism, organ-specific changes in lipid metabolism have also been reported. These changes are consistent with alterations in systemic metabolism; however, organ-specific changes can differ between tissues. In most tissues, the ability of organs to utilize lipids as energy substrates decreases, while lipids tend to accumulate. Several studies have shown that “ectopic lipid accumulation” occurs in various organs, including the liver, muscle, kidneys, and lungs [20,30].

This age-related lipid accumulation can be explained in several ways. First, aging alters lipid metabolism by regulating several important pathways involved in lipid transport. These include changes in adipose tissue lipolysis, lipoprotein and triglyceride metabolism, and changes in lipid transport proteins [31,32]. Second, aging is associated with changes in lipid metabolism inside the organs. Experimental and clinical data have shown that aging is associated with increased activity of lipid synthesis pathways with defective FAO [31,33,34]. In addition to these altered metabolic processes, changes in cellular organelles are also associated with aging and lipid metabolism [35]. Mitochondria play an essential role in lipid metabolism, and aging is associated with mitochondrial changes [36]. Although these changes are critical for maintaining cellular homeostasis, delineating the role of altered lipid metabolism on aging is difficult. Recent evidence suggests that these changes are not the passive results of aging, and that they contribute to accelerating the aging process (Figure 1).

### 2.1. Liver Lipid Accumulation and Aging

The liver is a main organ that mediates glucose, lipid, and amino acid metabolism. Liver lipid metabolism is associated with plasma lipid levels, hepatic uptake, the de novo synthesis of lipids, and the export of lipids from the liver [37]. These processes are tightly controlled under normal conditions, with the remaining minimum amounts of lipids in the liver. However, the liver easily accumulates lipids under various pathophysiological conditions [38]. Although acute lipid accumulation in the liver is reversible, as seen in the starvation-induced lipid accumulation model, chronic lipid accumulation is associated with various liver diseases [38]. Aging is also associated with considerable changes in lipid metabolism in the liver [39]. The incidence of non-alcoholic fatty liver disease (NAFLD) is steadily increasing in the elderly population, and has been demonstrated in aged animal models [40,41,42]. Advanced age leads to more severe histological changes in the liver and poorer clinical outcomes [39,43]. Progression of NAFLD is strongly associated with impaired systemic metabolism and metabolic syndrome, thereby contributing to age-related diseases.

The development of age-related NAFLD is associated with various factors that regulate liver lipid metabolism. The availability of lipid substrates usually increases with age [44]. However, it remains unclear whether there are changes in the hepatic uptake of lipids. Several fatty acid-binding proteins facilitate the transportation of fatty acids into hepatocytes, including fatty acid binding proteins (FABPs) and CD36. Changes in the expression of these proteins differ; while the levels of liver-type FABP decrease with aging, those of CD36 increase with aging [45,46]. This is associated with enhanced susceptibility to NAFLD progression. Changes in de novo lipogenesis have also detected in the aged liver. Multiple different enzymes are involved in de novo lipogenesis, including fatty acid synthase (FAS), acetyl-CoA carboxylase (ACC), and stearoyl-CoA desaturase (SCD) [47]. De novo lipogenesis is upregulated during aging, and transcriptional regulation by sterol regulatory element-binding protein 1 (SREBP1c) and carbohydrate-responsive element-binding protein (ChREBP) is thought to be important for this increase [39,48,49,50]. These transcription factors are mainly regulated by the nutrient sensing signaling pathway including mTOR, AMPK, and SIRT1, suggesting the importance of these factors in both aging and lipid metabolism. Experimental animal models have revealed the importance of de novo lipogenesis factors in the development of hepatosteatosis and insulin resistance [47,51].

In addition to uptake and lipogenesis, FAO is another important component of lipid metabolism. Mitochondrial β-oxidation is the major FAO pathway that metabolizes free fatty acids (FFAs) into acetyl-CoA. Key proteins involved in β-oxidation including carnitine palmitoyl transferase 1 α (Cpt1a) and acyl-CoA oxidase 1 (Acox1) are downregulated during aging [52,53]. Downregulation of FAO genes is explained by decreased transcriptional activity of peroxisome proliferator-activated receptor-alpha (PPARα) [54]. PPARα is major regulator of liver lipid metabolism which plays a critical role during the adaptive fasting response by promoting catabolism of fatty acids. The mRNA and protein expression of PPARα, and transcriptional activity of PPARα, were found to be reduced in aged livers [54]. Importantly, Howroyd et al. revealed a direct connection between PPARα and the aged liver. Mice lacking PPARα (whole body PPARα null mice) presented increased lipid accumulation with an overall decrease in liver function [55]. More recently, Montagner et al. generated hepatocyte-specific PPARα knockout mice and provided more direct evidence [56]. They found that liver PPARα is crucial for whole-body fatty acid homeostasis, and that it plays a protective role against NAFLD. In addition, they found that hepatocyte-restricted PPARα deletion is sufficient to promote NAFLD and hypercholesterolemia during aging. Park et al. showed that administration of a PPAR agonist decreases fatty liver in aging rats and delays the aging phenotype in the liver [57]. These results indicate that impaired PPARα in the liver plays an important role in aging-impaired liver lipid metabolism.

Autophagy, the degradation of cytosolic components by autophagosome formation, is another important mechanism for lipid degradation, termed lipophagy. The basal level of autophagy, as well as autophagy induced by stress responses, is impaired in the aged liver, making it vulnerable to liver damage [58,59,60]. Deficient autophagy is associated with increased lipid accumulation and impaired energy production in the aged liver [61]. Interestingly, PPARα regulates the transcription of autophagy-related genes [62]. Although a direct link between age-related lipid accumulation and PPARα in autophagy induction has not been investigated, it is plausible that decreased PPARα levels might contribute to lipid accumulation, partly, through deficient autophagy processes. Other major players involved in autophagy and autophagy-inducing signaling are also impaired in the aged liver including AMPK [63]. In addition to lipid degradation, liver autophagy recycles cellular organelles, including mitochondria [64]. Impaired mitophagy has been detected under pathological conditions in the liver, which further impairs normal lipid metabolism [65,66]. Taken together, this evidence indicates that autophagy is essential for maintaining lipid metabolism in the liver.

Mitochondrial integrity is also impaired in aged livers. Mitochondria play a key role in energy production via oxidative phosphorylation (OXPHOS) and fatty acid oxidation. Quality control of this organelle is critical for maintaining adequate metabolism; thus, impaired mitochondrial function is associated with several metabolic diseases. Changes in mitochondrial physiology have also been observed in the aged liver [67,68]. The number of mitochondria was found to decrease, concurrent with a decrease in mtDNA copy number and mitochondrial protein levels [67,69]. Mitochondrial dynamics (fission and fusion) and functional changes have also been detected in the aged liver, and these changes are associated with increased lipid accumulation [70]. In addition, impaired mitochondria are also associated with excessive oxidative stress, linking impaired metabolism to increased cellular stress [71]. This evidence suggests that changes in mitochondrial integrity during aging are related to lipid accumulation in the liver, leading to age-related liver diseases.

Taken together, these data indicate that various changes in lipid metabolism, such as lipid synthesis, lipid catabolism, and mitochondrial changes, occur in the aged liver. Although these changes differ between experimental species and models, it is clear that aging induces lipid accumulation in the liver. Liver lipid accumulation can lead to hepatic disorders, including steatohepatitis and cirrhosis, and impact systemic sequelae, including systemic glucose metabolism and metabolic syndromes.

### 2.2. Kidney Lipid Metabolism and Aging

Kidneys consume large amounts of energy to accomplish the reabsorption and secretion [72]. Although energy demand and substrate availability may vary by region, the basal energy consumption level is very high. Renal tubular epithelial cells have a high energetic demand; however, these cells have a relatively low glycolytic capacity [73,74,75]. Thus, renal tubular epithelial cells prefer FAO as a major energy source, and the high energy yield of FAO can support the high energetic demand [72]. In addition, kidney tubular epithelial cells possess a high number of mitochondria, in which FAO occurs [76]. Thus, maintaining renal lipid metabolism is important for ensuring that kidney cells produce an adequate amount of energy [77]. Supporting the importance of lipid metabolism, abnormal renal lipid metabolism has been observed both in chronic kidney disease (CKD) patients and in CKD animal models [78,79]. Details of these changes differ between models, but lipids tend to accumulate in both the tubule and glomerular regions [80,81]. This lipid accumulation is due to increased lipid synthesis and reduced FAO.

Aging is associated with changes in kidney function and structure. The effects of aging on the kidneys are most marked among other organs, because of the diverse factors that accelerate these changes. With aging, there is a progressive decline in renal function, including decreases in the glomerular filtration rate and increases in the urinary excretion of proteins, such as albumin. Similar to other CKD models, aged kidneys also present altered lipid metabolism leading to lipid accumulation [33,82,83]. Sun et al. showed that increased expression of SREBP transcription factors in the kidney resulted in glomerulosclerosis and proteinuria, suggesting the importance of lipid accumulation in kidney diseases [84]. Furthermore, same group observed a similar phenomenon in a model of aging. They found that with aging, triglycerides and cholesterol accumulate in the kidney, which is also a result of SREBP activation, suggesting the importance of renal lipid metabolism in age-related kidney diseases [83]. Anti-aging CR further decreased the expression of SREBP proteins, lipid accumulation, and the development of age-related renal diseases [82]. Collectively, these results demonstrate a role for increased lipid accumulation during kidney aging.

In addition to lipid synthesis, changes also occur in the FAO pathway under disease conditions. As mentioned above, because the kidney relies on the FAO pathway to produce energy, defective FAO pathways have been detected in most kidney disease models. Kang et al. showed that decreased PPARα expression is associated with defective FAO in renal tubular epithelial cells of diseased kidneys [72]. In addition, defective FAO in tubule epithelial cells leads to ATP depletion, cell death, and de-differentiation, contributing to the development of kidney disease [72]. Restoring FAO metabolism by genetic or pharmacological methods was shown to protect mice from tubulointerstitial fibrosis, suggesting that the maintenance of FAO plays an important role in kidney pathophysiology [72,85]. Defective FAO has also been detected during aging. Sung et al. showed that the protein level and transcriptional activity of PPARα were decreased in aged kidneys [86]. The decrease in FAO signaling is associated with lipid accumulation and fibrosis in aged kidneys. Critically, lipid accumulation and fibrosis were increased in mice lacking PPARα, and cortical cysts were detected during aging, suggesting that impaired PPARα and FAO accelerate the aging phenotype of the kidney [33,87]. Mitochondrial biology is also altered in aged kidneys. The number of mitochondria, and their structural changes, have also been reported in aged kidneys, in addition to changes in mitochondrial components [88,89]. Although there is no direct evidence that mitochondrial changes affect fatty acid metabolism during aging, it is plausible that defective mitochondria may be important for the dysregulation of lipid metabolism and kidney disease.

### 2.3. Skeletal Muscle Lipid Metabolism and Aging

Skeletal muscle is another important organ, which utilizes lipids and energy expenditure, and plays a central role in regulating whole-body energy homeostasis. Similar to other organs, muscle lipid metabolism includes the cycle of lipid uptake, synthesis, release, and energy generation [90]. However, the biology of lipid metabolism in skeletal muscle is somewhat complicated compared to that in other organs, because these processes are dependent on energy demand as well as substrate availability [90]. Nevertheless, defects in these processes lead to the accumulation of lipids in the muscle and have been linked to the development of metabolic disorders such as obesity and type 2 diabetes [91,92].

Sarcopenia, a type of muscle loss that occurs with aging, is accompanied by various physiological and biochemical changes in muscle biology. Among these, intermuscular lipid accumulation is common in skeletal muscle cells, and is usually accompanied by an increase in whole-body adiposity [93]. In both clinical and animal studies, lipid accumulation has been detected in the intermuscular region [93,94,95]. These changes are also associated with the loss of muscle quality and function. The mechanism of lipid accumulation in the muscle is similar to that in other organs. The imbalance between lipid uptake, lipid synthesis, and FAO is considered to affect lipid accumulation during muscle aging [96,97,98,99]. The changes in mitochondrial capacity to oxidize fatty acids is also important factor leading to the lipid accumulation in skeletal muscle [100,101].

The harmful effects of lipid accumulation can be explained in several ways. Previous studies have suggested a detrimental role of toxic lipid metabolites in sarcopenia [102]. Increases in toxic lipid metabolites in skeletal muscle are common during the development of sarcopenia, and evidence shows that toxic metabolites influence muscle cell biology in various ways. Several hypothesized mechanisms may synergistically drive lipotoxicity during aging and its functional consequences on muscle biology. These include de-differentiation of adipocyte-like progenitor cells, cellular senescence, and pro-inflammatory secretory factors [102]. Lipotoxicity with increased inflammation further alters the protein composition of single fibers, redox imbalance, and loss of regenerative capacity, leading to muscle dysfunction. The specific lipotoxic effects on muscle cells are covered extensively in other review articles; therefore, this review will not discuss this topic further [102].

More recently, defects in FAO or mitochondrial integrity have been shown to play important roles during sarcopenia development [103,104]. Muscle contains abundant mitochondria compared to other organs, and depends on FAO to produce energy when glucose is scarce. Therefore, it is plausible that defects in mitochondria or FAO signaling may play a significant role in muscle aging. Pollard et al. showed skeletal muscle mitochondria had a decreased abundance of phosphatidylethanolamine, but a pronounced increase of triglyceride levels. Reduced levels of phosphatidylethanolamines are known to decrease mitochondrial membrane fluidity and are connected with accelerated aging [105]. Henique et al. showed that increasing mitochondrial muscle FAO induces skeletal remodeling toward an oxidative phenotype [106]. Those authors utilized a mouse model that expressed constitutively active CPT1, a key FAO enzyme, in muscles. These mice presented increased FAO and oxidative fibers. In the context of aging, CPT1 expression in aged mice was shown to partially reverse aging-associated sarcopenia and fiber transition with increased muscle capillaries [106]. Shcherbakov et al. further demonstrated the importance of mitochondrial integrity in muscle aging [107]. They used a genetically altered mitochondrial mistranslation experimental model, which presented impaired mitochondrial function in muscle. Among the changes detected by RNA-sequencing and metabolomics analysis, they found that lipid metabolism and inflammation were increased in their model [107]. Those authors concluded that mitochondrial misreading in skeletal muscle accelerates metabolic aging, leading to lipid accumulation and increased inflammation, suggesting an important role for mitochondria and lipid metabolism in muscle aging. These changes were accompanied by increased glycolysis, lipid desaturation and eicosanoid biosynthesis, and alterations of the pentose phosphate pathway.

Although different in muscle type, cardiac muscle changes during aging is also associated with lipid metabolism in a similar way [108]. Balanced cardiac lipid metabolism is essential for normal function of the heart, and reduced fatty acid metabolism may be detrimental for cardiac function [109]. Aging-related cardiomyopathy has been associated with downregulation of PPARα with impaired mitochondrial function similar with skeletal muscle aging [109,110]. Together, recent studies have shown that maintaining adequate FAO and mitochondrial integrity is crucial for muscle function during aging.

## 3. Role of Lipid Accumulation in Adipose Tissue: Adipokines on Systemic Aging

Adipose tissue is a central organ that is specialized for fat storage. Adipose tissue is highly dynamic and capable of continuously changing size according to the amount of energy in the body [111]. Two major types of adipose tissue exist: white adipose tissue (WAT) and brown adipose tissue (BAT). The key physiological functions of WAT include insulation and energy storage [111]. Among the subtypes of WAT, visceral WAT mainly stores dietary lipids or excessive calories in the form of triglycerides. WAT releases the stored energy in response to caloric deficits to provide energy for physiological functions in the body. BAT is a specialized form of adipose tissue, which participates in non-shivering thermogenesis through lipid oxidation [112]. These adipose tissues were once considered to have passive roles in energy storage and mobilization. However, adipose tissue is an active endocrine organ that influences various physiological processes throughout the body [113]. These systemic effects of adipose tissue are due to adipokines, which are cytokine-like molecules secreted by adipose tissue [113]. The first adipokine discovered was leptin, and hundreds of adipokines have since been identified [114]. The amount of adipokine secretion depends on the condition of adipose tissue. The secretion of adipokines is particularly altered under conditions of obesity, contributing to metabolic disorders (Figure 2).

### 3.1. Changes in Adipose Tissue during Aging

Similar to other organs, age-related changes occur in the adipose tissue. Interestingly, obesity and aging share similar mechanisms and effects on adipose tissue [115]. Because adipose tissue dysfunction has systemic effects, adipose tissue has been an interesting target in aging intervention studies. Aging is associated with significant changes in adipose tissue biology [116]. Inside the fat tissue, changes in abundance, distribution, and cellular composition have been observed [117]. These changes are related to the endocrine function of adipose tissue, and further influence the development of insulin resistance and metabolic dysfunction [117]. Although there are sex differences in adipose tissue deposition with aging, body mass and body fat percentage increase in both males and females during aging [118]. A notable change in adipose tissue during aging is fat redistribution. The main redistribution involves the shifting of adipose tissue fat from subcutaneous to visceral depots [119,120]. Subcutaneous and visceral adipose depots differ in terms of the systemic effects. Accumulation of fat in the visceral tissue is associated with an increased risk of metabolic syndrome, while subcutaneous fat accumulation is associated with a lower risk of disease [121,122,123].

In addition to fat redistribution, cellular changes have also been reported during aging. Adipose tissue is composed of various cell types and is divided into two fractions. The adipocyte fraction contains primarily mature adipocytes, and the stromovascular fraction contains progenitor cells, lymphocytes, endothelial cells, pericytes, and fibroblasts. The function and adipogenic potential of aged adipose progenitor cells are reduced compared to those of young progenitor cells [117,124]. In addition, pre-adipocytes present reduced insulin responsiveness, which limits the ability of adipose tissue [125]. Impaired plasticity of progenitor cells and pre-adipocytes leads to a functional decline of adipose tissue and limits its ability to expand under conditions of excess nutrients. Cellular senescence in adipose tissue is another feature of aging [126]. In both aging and obesity, considerable senescent cells accumulate in adipose tissue [127]. Compared to normal cells, senescent cells exhibit different phenotypes, including increased expression of cell cycle arrest-related proteins, membrane proteins, and senescence-associated β-galactosidase (SA-β-gal), which affect cellular function [117]. These cells alter the ability of adipose tissue, and influence systemic aging via the senescence-associated secretory phenotype (SASP). The SASP involves various cytokines, chemokines, growth factors, and matrix metalloproteinases. The systemic circulation of SASP is closely associated with inflammation, which contributes to aging and age-associated chronic diseases [128]. The exact mechanism through which SASP contributes to aging has been discussed in several recent reviews [129,130].

Along with senescent cells, inflammatory cells in adipose tissue secrete cytokines and chemokines that can circulate throughout the body. Adipose tissue macrophages (ATMs) are a significant source of circulating cytokines, including TNFα, IL-1β, and IL-6 [131]. Interestingly, the number of ATMs differs between visceral and subcutaneous fat. With age, ATMs accumulate in subcutaneous fat, but no significant change has been detected in visceral depots, suggesting that subcutaneous fat is a sentinel source of inflammatory cytokines [132,133,134]. However, although there is no change in the number of ATMs, studies have shown that the pro-inflammatory M1 macrophage type dominates in aged visceral adipose tissue, and this may also contribute to systemic inflammation [134]. Other immune cells in the adipose tissue change during aging. Most T cells, including CD4+ lymphocytes and CD8+ lymphocytes, accumulate in aged visceral adipose tissue, contributing to regional and systemic inflammation [134,135]. Thus, it is evident that the pro-inflammatory phenotype of aged adipose tissue changes adipose tissue itself and also increases systemic inflammation leading to aging and age-related diseases.

### 3.2. Role of Adipokines in Aging: Leptin and Adiponectin

Leptin was the first adipokine to be discovered, in the 1990s. Leptin is released in response to changes in nutritional status [114]. Following food intake, leptin is released from adipose tissue and inhibits appetite by regulating neural circuits in the brain. Thus, adipose tissue as an endocrine organ actively involves in modulating energy metabolism and homeostasis [136]. Leptin or leptin receptor-deficient mice are obese because of their uncontrolled appetite and excessive food intake. In addition, leptin also increases lipid usage in peripheral tissues and promotes mitochondrial biogenesis, which accelerates the overall energy expenditure by the body [114]. Circulating leptin levels are elevated along with increased adipose tissue mass under conditions of obesity; however, like insulin resistance, leptin resistance aggravates metabolic disease due to inadequate appetite and metabolic control [137]. Leptin levels have been shown to increase with aging in experimental animals and humans [138]. However, because leptin levels are correlated with total fat mass, which increases with aging, this suggests that age does not have an independent effect on leptin expression. In addition, diminished leptin sensitivity has been detected during aging [138,139]. Visceral fat was not decreased in older rats administered leptin, as observed in young rats, and leptin gene expression was not suppressed in adipose tissue. This may be due to decreased receptor expression or changes in leptin signaling in leptin-responsive cells.

Adiponectin is another adipokine expressed in adipose tissue [140]. In contrast to leptin, adiponectin expression is negatively correlated with fat mass. AdipoR1 and AdipoR2, receptors for adiponectin, are widely distributed throughout the body [140]. Activated receptors exert anti-obesity and antidiabetic effects and alleviate insulin resistance by stimulating AMP-activated protein kinase (AMPK) signaling [141]. Adiponectin also inhibits pro-inflammatory cytokine production by blocking NF-κB activation [142]. Interestingly, serum adiponectin levels are elevated during aging, regardless of changes in fat mass [143]. The level of adiponectin was found to be positively correlated with the extension of longevity [144]. For example, centenarians have higher levels of adiponectin, which may be associated with extended longevity [144]. However, increased adiponectin levels during aging also have negative effects. This is termed the “adiponectin paradox in the elderly.” Regardless of its positive role, greater serum adiponectin levels are associated with low skeletal muscle, low muscle density, and poor physical function [145]. High adiponectin levels are also associated with a greater risk of incident disability and death. More research is needed to determine whether modulation of these adipokines could have a positive effect on aging or age-related metabolic diseases.

### 3.3. Evidences Suggesting Active Role of Adipose Tissue in Aging

Based on its systemic importance, adipose tissue has been suggested to play an integral role in longevity. Several studies have shown that interventions aiming to extend lifespan affect the biology of adipose tissue. Anti-aging interventions, including dietary restriction models (CR), gene mutation models (e.g., in the GH/IGF-1 axis), and pharmacologic intervention models (metformin, resveratrol), regulate fat tissue, either directly or indirectly [146,147,148,149]. In these models, less fat mass is detected in adipose tissue through pathways related to nutrient and hormonal signaling. In addition, lower levels of ectopic lipid accumulation have been observed in metabolic organs, suggesting the systemic importance of lipid metabolism. CR models, which have extended the longevity of most species, present marked changes in fat mass and adipose tissue biology [150]. Within the adipose tissue, lower levels of inflammation and cellular senescence have been reported, while tissue protective mechanisms (autophagy, DNA repair processes, and organelle integrity) were found to be upregulated [151]. In a growth hormone-deficient mouse model, adipose tissue improved adipose progenitor function with reduced senescent cells [117,147]. Metformin, which induces a small but significant lifespan extension in rodents, reduces body mass in both rodent models and humans mainly through a decrease in adipose tissue [148,152]. Resveratrol, another pharmacological anti-aging candidate, has also been shown to exert changes in adipose tissue biology. Interestingly, resveratrol affects adipokines involved in the fat browning process, leading to the maintenance of a healthy weight against obesity [153]. Collectively, these data suggest that anti-aging strategies are associated with changes in adipose tissue biology.

In contrast, evidence also supports the anti-aging effects of adipose-tissue specific interventions. Bluher et al. first introduced fat-specific insulin receptor knockout (FIRKO) mice, which have less fat mass with longevity extension, similar to whole-body insulin receptor knockout mice [154]. Follow-up studies from the same research group showed that maintaining mitochondrial activity and metabolic rate in adipose tissue may be important for increasing the lifespan of FIRKO mice [155]. Similarly, adipose tissue-specific p110α (PI3K)-deleted mice also presents reduced fat accumulation over a significant part of their older life and allowed the maintenance of normal glucose tolerance despite insulin resistance [156]. They identified a potentiating effect on β-adrenergic signaling, leading to increased catecholamine-induced energy expenditure in adipose tissue. These results suggest that the regulation of insulin signaling in adipose tissue is sufficient for lifespan extension, implying the importance of adipose tissue in aging processes. In addition to the genetic ablation model, surgical removal of adipose tissue has also demonstrated a life-extension effect. The removal of visceral fat prevented insulin resistance and glucose intolerance in aged rats, and increased the median and maximum lifespan [157]. Removal of omental adipose tissue also demonstrated beneficial effects on insulin sensitivity in healthy dogs [158]. However, no clear evidence has been reported with regards to human aging. Experiments to remove omental adipose have been conducted in obese, diabetic individuals undergoing gastric bypass surgery, and no additional beneficial effects on metabolic health were reported [159]. More research is needed to determine whether the removal of adipose tissue is beneficial for humans.

## 4. Other Important Aspects of Lipid Metabolism in Aging

### 4.1. Changes in Lipid Metabolites and Their Implication in Aging

In addition to the classical role of lipid metabolites as energy sources, several lipid metabolites have other functions, including a role in signaling pathways and as structural components of cell membranes. These functions usually involve specific forms of fatty acids or lipid metabolites, other than fatty acids. Several polyunsaturated fatty acids (PUFAs) play essential roles in cellular signaling pathways [160]. Arachidonic acid, derived from dietary linoleic acids, is a precursor of important eicosanoids that control a wide array of body functions including inflammation, vasodilation, and cell growth. Other essential PUFAs, including omega-3 and omega-6, have important roles in various cellular processes [161]. Omega-3 fatty acids have strong anti-inflammatory effects by inducing resolution phase of inflammation. Phospholipids are key components of the lipid bilayer of cells, which maintain the integrity of the cell membrane. Some phospholipids are also involved in cellular signaling pathways [162]. Sphingolipids, which are similar to phospholipids, participate in maintaining plasma lipid bilayers. Specific sphingolipids, such as sphingosine, are important mediators in apoptosis, proliferation, stress response, necrosis, and inflammation pathways [163]. Furthermore, ketone bodies derived from fatty acids, which serve as a circulating energy source for tissues during fasting or prolonged exercise, are also involved in cellular signaling pathways [164]. Cholesterol and lipoprotein are important lipid-derived factors that participate in various physiological processes [165,166]. Cholesterol binds to and affects various ion channels, while it also activates the estrogen-related receptor alpha (ERRα), and may be the endogenous ligand for the receptor. The implications their contributions to disease development are well described.

Based on their functions, the effects of specific lipid metabolites on aging, lifespan regulation, and age-related diseases have been widely studied. These approaches include lipidomic analysis of young and aged individuals, genetic alterations in lipid metabolism-associated tissue, and the direct administration of specific lipid metabolites [24]. Because Johnson and Stolzing comprehensively reviewed these topics, a few recent examples showing the lifespan-extending effect of lipid-related interventions will be discussed here [24]. Several lipid-related genetic interventions have extended the lifespan of *Caenorhabditis elegans*. Knockdown of the ceramide synthase gene, hyl-1, can extend life [167]. Depletion of fatty acid desaturase fat-4 significantly extended life and increased resistance to oxidative stress [168]. Interestingly, Han et al., linked chromatin remodeling to lipid metabolism during aging in *C. elegans* [169]. They found that H3K4me3-methyltransferase deficiency, which extends lifespan, promotes fat accumulation with a specific enrichment of mono-unsaturated fatty acids (MUFAs). Dietary MUFAs or the overexpression of fat-7 in the intestine, are sufficient to extend the lifespan [169]. In fruit flies and mice, overexpression of apolipoprotein D increased the lifespan and enhanced resistance to oxidative stress [170,171]. Increased apolipoprotein D also reduced age-associated lipid peroxide accumulation. Lipid-related dietary interventions that extend lifespan have also been reported. In *C. elegans*, administration of α-lipoic acid, omega-3 lipids, β-hydroxybutyrate (ketone body), MUFAs, and α-linolenic acid has been shown to increase lifespan [169,172,173,174,175]. Treatment with steroid 17-α-estradiol extended the lifespan of male mice fed a ketogenic diet also presented an increased lifespan with improved motor function and memory, muscle mass, and function [176,177,178]. These examples demonstrate how specific metabolites derived from lipids play a role in aging processes.

### 4.2. Epigenetic, Lipid Metabolism, and Aging

One possible mechanism through which lipid metabolism is regulated during aging, or how aging changes lipid metabolism, is epigenetic regulation. Recent evidence suggests that changes in lipid metabolism can alter the lifespan by regulating chromatin states [179]. Chromatin modifications, such as changes in histone and DNA, regulate gene expression, especially under chronic conditions, such as aging [180]. Indeed, the state of chromatin marks alters with age, and several chromatin modifiers influence lifespan in a variety of species [181]. Changes in lipid metabolism are associated with chromatin regulation. Several studies have shown that specific lipids or lipid metabolites directly act on histone acetylation and acylation. Acetyl-CoA, which is a product of FAO, is used as a cofactor to add acetyl groups to the lysine residue [182]. Short-chain fatty acids are also involved in histone modification [183,184]. In addition, lipid metabolism and chromatin modification share common precursors, including S-adenosyl methionine (SAM). SAM is required for the generation of phosphocholines in phospholipids, while histone methylation is also dependent on SAM [185]. There is also evidence that lipid-induced changes in signaling pathways indirectly regulate chromatin states. Pro-inflammatory lipid molecules, including lipopolysaccharides and eicosanoids, have been suggested as potent epigenetic modifiers [162,186]. A variety of lipids that can bind GPCRs, including free fatty acids, their derivatives, and phospholipid derivatives, also modulate the epigenomic landscape by inducing cellular signaling [187]. However, there is no direct evidence that answers whether lipids impact lifespan by affecting chromatin marks, further studies are necessary to reveal the importance of their interactions.

On the other hand, changes in chromatin modifications can affect lipid metabolism and lifespan. Histone methyltransferases have been shown to link epigenetic modifications to lipid metabolism. In *C. elegans*, a deficiency in H3K4me3 modifiers extends lifespan [169]. These changes were associated with MUFA synthesis, and inhibition of MUFA synthesis was sufficient to reduce lifespan extension. These findings link changes in histone modification to lipid metabolism, especially MUFAs, and their role in lifespan extension [169]. Sirtuin histone deacetylases are chromatin modifiers that link metabolism and aging [181]. By deacetylating histone and non-histone proteins, sirtuins regulate important metabolic processes, including lipid metabolism and longevity [188]. DNA methylation also influences lipid metabolism. The results of recent studies have shown the importance of DNA methylation in lipid metabolism and aging. Hahn et al. evaluated changes in DNA methylation induced by dietary restriction (DR) [189]. By profiling genome-wide changes in DNA methylation, gene expression, and lipidomics in response to DR and aging in mouse liver, those authors found that changes at loci involved in lipid metabolism affect gene expression and consequently, lipid profile. More recently, Li et al. showed that impaired lipid metabolism due to age-dependent changes in DNA methylation was associated with aging [190]. In addition, they showed that elongation of very long chain fatty acid elongase 2 (Elovl2), a gene whose epigenetic alterations are most highly correlated with age prediction, contributed to aging by regulating lipid metabolism. This evidence supports a role for epigenetic modification in bridging lipid metabolism and aging.

### 4.3. Other Lifespan-Extending Interventions that Affect Lipid Metabolism

Several studies have identified an indirect association between lipid metabolism and aging. Canaan et al. showed that knockout of the ubiquitin-like gene FAT10 extended the lifespan of mice [191]. Although FAT10 is not directly associated with lipid metabolism, knockout mice have an increased metabolic rate with markedly reduced adiposity and inflammation. Griveau et al. demonstrated the effect of the phospholipase A2 receptor (Pla2r1) on regulating lifespan in a mouse model of progeria [192]. Whole-body knockout of Pla2r1 decreased premature aging phenotypes, together with decreased senescence markers. Yoshida et al., showed that adipose tissue-specific overexpression of nicotineamide phosphoribosyl transferase (NAMPT) increased NAD+ levels in multiple tissues, which further extended the lifespan of mice [193].

## 5. Concluding Remarks

In conclusion, lipid metabolism plays an essential role in regulating the aging process. Experimental evidence shows that lipid metabolism is changed during aging and lipid-related interventions can modulate age-related diseases and aging in various model organisms. In addition to systemic changes in lipid metabolism, tissue-specific metabolic changes have been widely investigated with advances in genetic engineering techniques. Critically, recent studies have implicated deficient FAO processes as a cause of dysregulated lipid metabolism during aging. FAO deficiency increases ectopic lipid accumulation and affects various cellular processes that contribute to the aging process. Age-related mitochondrial changes are also associated with deficient lipid catabolism. In adipose tissue aging, the production of various adipokines has important implications in the regulation of systemic metabolic changes and aging processes. Moreover, specific lipids and lipid-derived molecules have been shown to increase or decrease in an age-dependent manner. The administration of specific lipid or lipid-derived molecules impacts biological processes, further influencing the lifespan of various species. Changes in lipid metabolism have also been associated with other important aspects of aging, including epigenetics. Overall, these evidences demonstrate that lipid metabolism is not a passive metabolic process, but has an active role in biological processes and contributes to aging and age-related diseases.

Although evidences suggest the link between lipid metabolism and longevity, additional studies are required to answer the remaining questions. In particular, future work should aim to better understand the mechanisms how lipid-related interventions extend lifespan in model organisms. Since most of the studies were done in non-vertebrate model organism including *C. elegans*, additional research in vertebrate model is required to further prove the importance of lipid metabolism in aging. Using advanced genetic alteration techniques such as conditional knockout or overexpression systems will be necessary to demonstrate specific role of lipid metabolism related signaling in the aging process. Another approach would be to do comprehensive analysis of lipidomics in animals with extreme longevity or in centenarians to reveal the lipids that is associated with longevity. While sophisticated studies are required to prove the importance of lipid metabolism changes during aging, lipid-related interventions may provide effective options for delaying the aging process and for the treatment of age-associated diseases.

## Figures and Tables

**Figure 1 cells-10-00880-f001:**
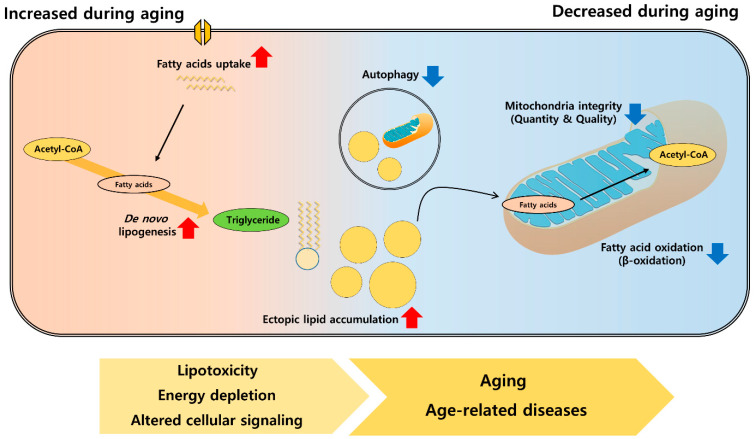
Age-related changes in lipid metabolism and their effects on aging and age-related diseases. Ectopic lipid accumulation occurring during aging is mainly induced by increased fatty acids uptake, de novo lipogenesis, with decreased fatty acid oxidation process. Mitochondrial integrity and autophagic induction is also diminished during aging leading to decreased lipid catabolism. These changes further bring lipotoxicity in the cell, deplete energy in the tissue, and alter cellular signaling causing accelerated aging and early onset of age-related diseases.

**Figure 2 cells-10-00880-f002:**
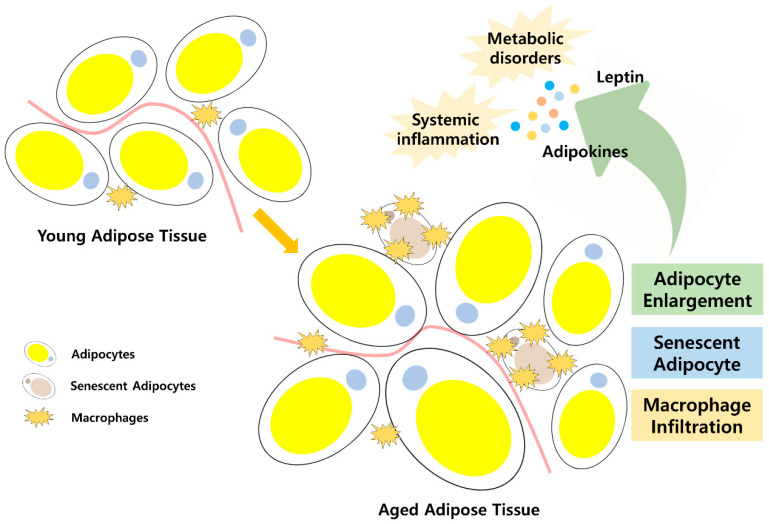
Changes in adipose tissue during aging and their effects to systemic aging. Compared to young adipose tissue, aged adipose tissue has enlarged, senescent adipocyte phenotype causing changes in adipokine secretion. Increase in inflammatory cells in adipose tissue secretes cytokines and chemokines that can circulate throughout the body. These changes during aging causes systemic inflammation and metabolic disorders.

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
