# Peer review of "Advances in Understanding of the Role of Lipid Metabolism in Aging"

_cells, 2021, doi:10.3390/cells10040880_

Round 1

Reviewer 1 Report

The submitted manuscript focuses on changes in lipid metabolism during aging with emphasis on ectopic lipid accumulation in the liver, kidney, and skeletal muscle as well as the role of adipokines produced by adipose tissue in aging. Summarizing the recent data on the role of alterations in lipid metabolism in aging could contribute to understanding the molecular mechanism of aging and the discovery of novel biomarkers and therapeutic agents to combat aging and age-related chronic diseases. However, there are some concerns and recommendations regarding the scientific quality and content of the manuscript.

Major concerns:

  1. The manuscript style is not equable throughout the entire manuscript. For example, section 2 “Evidence linking lipid metabolism to aging” is poorly written; it is superficial and contains many general and repeated sentences. While section 2.1 “Liver lipid accumulation and aging” is written somewhat better and contains a more detailed discussion of experimental data. Some sections contain excessive references to review articles. This especially concerns references to old papers such as ref [22].
  2. The manuscript focuses mostly on fatty acids and fatty acid oxidation (FAO). However, alterations in the metabolism of triglycerides, phospholipids, and other complex lipids were not considered. Also, there has been very little discussion of cellular and molecular mechanisms underlying dysregulation of lipid metabolism including cell signaling pathways in aging.
  3. It would be useful to give a schematic representation of metabolic pathways of lipid degradation and biosynthesis, their cross-links to glucose and amino acid metabolism. This is especially important since the author mentioned that FAO gives rise to acetyl-CoA, which can further enter the Krebs cycle to produce more ATP amount or can be used for ketogenesis (in starvation) and fatty acid biosynthesis (in excess energy conditions). Alpha-ketoacids produced in Krebs cycle can be used for amino acid biosynthesis, so on. Such a scheme can be amended by regulatory molecules with an indication of their dysfunction in aging.
  4. The work is not well-organized. The author discussed lipid accumulation in the liver, kidney, and muscles in three different subsections. However, each subsection discusses changes in the same process such as FAO, PPARα, and mitochondrial dysfunction. Instead, the author could discuss changes in membrane lipids, vascular lipids, lipoprotein oxidation, etc., and age-related chronic diseases caused by lipid metabolism dysfunction such as atherosclerosis, etc. Additionally, it would be useful to discuss genetic mutations associated with lipid metabolism in aging.

Other concerns:

  1. Lines 25-26 and 29-30: it is better to give a more complete definition of aging and to list all hallmarks of aging since the author stated that there are 9 hallmarks.
  2. The term “metabolic organs” is not fully correct since each organ has a unique metabolic profile. It is better to avoid this term.
  3. There many general sentences, which require more details. For example, on lines 195-196 “Other major players involved in autophagy and autophagy-inducing signaling are also impaired in the aged liver”. Which players?
  4. Titles of subsections 3.1 and 3.3 are very similar.
  5. Some repeated sentences should be removed.
  6. Other recommendations are given as Notes immediately in the .pdf file of the manuscript.

Author Response

  1. The manuscript style is not equable throughout the entire manuscript. For example, section 2 “Evidence linking lipid metabolism to aging” is poorly written; it is superficial and contains many general and repeated sentences. While section 2.1 “Liver lipid accumulation and aging” is written somewhat better and contains a more detailed discussion of experimental data. Some sections contain excessive references to review articles. This especially concerns references to old papers such as ref [22].

Response to reviewers: Thanks for your thoughtful suggestion. I agree with your insightful opinion. In the first part of section 2, I try to give general information (especially focused on systemic changes) before giving detailed information of each organ, thus contains somewhat superficial information. However, I also felt this section is not well organized, I made changes in the composition. Because the first part of section2 plays is the opening of section2, I shortened the overall explanation. I also replaced references.

  1. The manuscript focuses mostly on fatty acids and fatty acid oxidation (FAO). However, alterations in the metabolism of triglycerides, phospholipids, and other complex lipids were not considered. Also, there has been very little discussion of cellular and molecular mechanisms underlying dysregulation of lipid metabolism including cell signaling pathways in aging.

Response to reviewers: Thanks for your thoughtful suggestion. I agree with your insightful opinion. I also agree that alterations in the metabolism of triglycerides, phospholipids, and other complex lipids are important during aging. Due to shortage of permitted pages, it was not able to cover all the aspect of lipid metabolism. In the abstract, I mentioned that this review will be focused on recent findings with limited subjects. And I also mentioned and added several other review articles that cover broad ranges of lipid metabolism changes during aging. I also added several sentence that discusses cellular and molecular mechanisms underlying dysregulation of lipid metabolism.

  1. It would be useful to give a schematic representation of metabolic pathways of lipid degradation and biosynthesis, their cross-links to glucose and amino acid metabolism. This is especially important since the author mentioned that FAO gives rise to acetyl-CoA, which can further enter the Krebs cycle to produce more ATP amount or can be used for ketogenesis (in starvation) and fatty acid biosynthesis (in excess energy conditions). Alpha-ketoacids produced in Krebs cycle can be used for amino acid biosynthesis, so on. Such a scheme can be amended by regulatory molecules with an indication of their dysfunction in aging.

Response to reviewers: Thanks for your thoughtful suggestion. I agree with your insightful opinion. The point you mentioned is really important since lipid metabolism is also associated with other metabolic process as you mentioned. It will be really interesting to look at how acetyl CoA contribute to ketogenesis and amino acid synthesis. However, as I mentioned earlier, this review only focuses only limited topics.

  1. The work is not well-organized. The author discussed lipid accumulation in the liver, kidney, and muscles in three different subsections. However, each subsection discusses changes in the same process such as FAO, PPARα, and mitochondrial dysfunction. Instead, the author could discuss changes in membrane lipids, vascular lipids, lipoprotein oxidation, etc., and age-related chronic diseases caused by lipid metabolism dysfunction such as atherosclerosis, etc. Additionally, it would be useful to discuss genetic mutations associated with lipid metabolism in aging.

Response to reviewers: Thanks for your thoughtful suggestion. I agree with your insightful opinion. Again with the second question, due to shortage of permitted pages, it was not able to cover all the aspect of lipid metabolism. I try to add more information of lipid accumulation evidences in the manuscripts. The link between lipid metabolism and age-related diseases is also important. I am now preparing another review focused on the age-related diseases. Here, I just focused only on aging rather than age-related diseases.

Other concerns:

  1. Lines 25-26 and 29-30: it is better to give a more complete definition of aging and to list all hallmarks of aging since the author stated that there are 9 hallmarks.

    Response to reviewers: Thanks for your comments. I made change according to your suggestion.

  2. The term “metabolic organs” is not fully correct since each organ has a unique metabolic profile. It is better to avoid this term.

    Response to reviewers: Thanks for your comments. I made change according to your suggestion.

  3. There many general sentences, which require more details. For example, on lines 195-196 “Other major players involved in autophagy and autophagy-inducing signaling are also impaired in the aged liver”. Which players?

    Response to reviewers: Thanks for your comments. I made change according to your suggestion.

  4. Titles of subsections 3.1 and 3.3 are very similar.

    Response to reviewers: Thanks for your comments. I made change according to your suggestion.

  5. Some repeated sentences should be removed.

    Response to reviewers: Thanks for your comments. I made change according to your suggestion.

  6. Other recommendations are given as Notes immediately in the .pdf file of the manuscript.

    Response to reviewers: Thanks for detailed comments on this manuscripts. Your review was really helpful to increase manuscript quality and even gave me new insight with this topic.

Reviewer 2 Report

In this work Chung strives to describe advances in the understanding of contribution of lipid metabolism to aging.  The Author cites 169 articles out of which 37% were published within last 5 years and 11% were published within last 2 years. In general the article is well planned, the outline is described in the introduction section and the main aim of the paper is clearly stated. However there are several points that needs to be addressed prior to publication:

  • Some parts of the article contains very general, elusive information without concreate data. While those are justified in the section such as Introduction or conclusions they should be avoided elsewhere. In some cases the Author just touches the subject without giving any specific information. For example:
    1. In chapter 2 the Author states that “aging alters lipid metabolism by regulating several important pathways” – what are those pathways?
    2. Then, the Author states that “changes [in mitochondria] increase ectopic fat accumulation but also lead to other problems associated with aging” – what problems?
    3. Further, the Author states that “age-induced reduction in lipolysis is associated with decreased availability of catecholamines..” but doesn’t explain how this is associated with lipolysis.
    4. In chapter 2.1 the Author states that “key proteins involved in beta-oxidation are downregulated during aging” – what proteins exactly?
    5. In the 1st paragraph of chapter 4.1 the author lists some groups of lipids and describes their function as ‘important role in various cellular processes’ or ‘participate in various cellular processes’ or ‘control wide array of body functions’ – use of such phrases doesn’t convey meaningful information and could be synthesized into one or two sentences. Alternatively, (and preferably) more details can be added explaining exact cellular processes and functions that described lipids are involved in.
  • In the chapter 4.2 the Author demonstrates that lipid metabolism is associated with chromatin regulation and is necessary for epigenetic changes, but doesn’t explain how it changes and how it may contribute to aging. Is such data available?
  • In the last chapter (chapter 5) the Author tries to synthesize the information included in the article. However, this recap fails to convince the reader that “lipid metabolism is not a passive metabolic process, but has an active role in… aging and age related diseases”. This part of the article needs to be re-structured in a way to clearly summarize how lipid metabolism actively contributes to aging. Additionally, the Author mentions that “more studies are required to answer the remaining questions..” – what questions the Autor has in mind? It is worth stating them here and maybe shortly explain what are/should be (according to the Author) the future directions of studies on lipid metabolism.

Minor points:

  • I find the phrase “extend longevity” a bit awkward. Usually we talk about extending lifespan, which I would rather use here.

Author Response

In this work Chung strives to describe advances in the understanding of contribution of lipid metabolism to aging. The Author cites 169 articles out of which 37% were published within last 5 years and 11% were published within last 2 years. In general the article is well planned, the outline is described in the introduction section and the main aim of the paper is clearly stated. However there are several points that needs to be addressed prior to publication:

Response to reviewers: Thanks for the detailed comments on the manuscripts. The reviewer’s suggestion was really helpful to improve the article.

Some parts of the article contains very general, elusive information without concreate data. While those are justified in the section such as Introduction or conclusions they should be avoided elsewhere. In some cases the Author just touches the subject without giving any specific information. For example:

In chapter 2 the Author states that “aging alters lipid metabolism by regulating several important pathways” – what are those pathways?

Response to reviewers: Thanks for the pointing out. I tried to change the introduction part of chapter 2. The first part of chapter 2, I try to give general information (especially focused on systemic changes) before giving detailed information of each organ, thus contains somewhat superficial information. Following the statement that reviewer pointed out, the next statement “These include changes in adipose tissue lipolysis, lipoprotein and triglyceride metabolism, and changes in lipid transport proteins” meant several important pathways. In the following chapters (2.1, 2.2, and 2.3), I tried to give specific information for each organ.

Then, the Author states that “changes [in mitochondria] increase ectopic fat accumulation but also lead to other problems associated with aging” – what problems?

Response to reviewers: Thanks for the pointing out. I reorganized the introductory part of chapter 2 to avoid such confusion.

Further, the Author states that “age-induced reduction in lipolysis is associated with decreased availability of catecholamines..” but doesn’t explain how this is associated with lipolysis.

Response to reviewers: Thanks for the pointing out. I added explanation how reduction in lipolysis is associated with decreased availability of catecholamines.

In chapter 2.1 the Author states that “key proteins involved in beta-oxidation are downregulated during aging” – what proteins exactly?

Response to reviewers: Thanks for the pointing out. I added more information after the statement.

In the 1st paragraph of chapter 4.1 the author lists some groups of lipids and describes their function as ‘important role in various cellular processes’ or ‘participate in various cellular processes’ or ‘control wide array of body functions’ – use of such phrases doesn’t convey meaningful information and could be synthesized into one or two sentences. Alternatively, (and preferably) more details can be added explaining exact cellular processes and functions that described lipids are involved in.

Response to reviewers: Thanks for the thoughtful suggestion. I agree with your opinion that some of the phrases lack detailed information. I tried to add more information for each lipid. However, the review did not focus to explain all the lipid species, it was not able to add whole information for each lipid.

In the chapter 4.2 the Author demonstrates that lipid metabolism is associated with chromatin regulation and is necessary for epigenetic changes, but doesn’t explain how it changes and how it may contribute to aging. Is such data available?

Response to reviewers: Thanks for the insightful point. I agree with your opinion that there is no direct evidence that answers whether lipids impact lifespan by affecting chromatin marks. Because many lipid species can play as epigenetic modifier, and epigenetics actually plays important role in the aging process, it is possible that lipids can extend lifespan through epigenetic modification. I still think further studies are necessary to reveal the importance of their interactions. I added some phrases on that topic in the chapter 4.2 for clear explanation.

In the last chapter (chapter 5) the Author tries to synthesize the information included in the article. However, this recap fails to convince the reader that “lipid metabolism is not a passive metabolic process, but has an active role in… aging and age related diseases”. This part of the article needs to be re-structured in a way to clearly summarize how lipid metabolism actively contributes to aging. Additionally, the Author mentions that “more studies are required to answer the remaining questions..” – what questions the Autor has in mind? It is worth stating them here and maybe shortly explain what are/should be (according to the Author) the future directions of studies on lipid metabolism.

Response to reviewers: Thanks for the insightful point. I agree with your opinion that the last chapter failed to convince the idea. I try to reorganize the section with adding opinions for the future directions of studies on lipid metabolism.

Minor points:

I find the phrase “extend longevity” a bit awkward. Usually we talk about extending lifespan, which I would rather use here.

Response to reviewers: I changed the word.

Reviewer 3 Report

This manuscript (MS) reviewed the current understanding of age-related change in lipid metabolism that may contribute to age-related diseases. It focused on the epigenetic changes and endocrine features of lipid metabolism during aging. Generally, it is ok for summarizing recent research program. However, there are several point concerning this article for its improvement:

  1. As one of the ultimate purposes of the research of age-related lipid metabolism is to understand its influence in multiple age-related diseases, more of its links with diseases are supposed to be described. The following includes some reviews relating to this topic. These or other similar publications are recommended to be involved in the manuscript with brief descriptions on the mechanisms underlying age-related lipid metabolism and age-related diseases.

  • age-related lipid metabolism AND Alzheimer’s disease:

Reference-1: Int J Mol Sci. 2020 Feb; 21(4): 1505. Lipids and Alzheimer’s Disease. PMID: 32098382.

Refernce-2: Front Aging Neurosci. 2015 Oct 23; PMID: 26557086

  • age-related lipid metabolism AND Diabetes:

Reference: Lancet . 2010 Jun 26;375(9733):2267-77. Lipid-induced insulin resistance: unravelling the mechanism.PMID: 20609972

  1. Line 386 to 388 is about the T cells accumulation at aged adipose tissue contributing to regional and systemic inflammation. There are two points needed to be modified:

Firstly, references supporting this statement are supposed to be given. In aged immune system, there is an increased population of senescent T cells that contributing inflammation, therefore, the supporting reference here should include senescent T cell accumulation in aged adipose tissue. The following is one example reference about PD-1+ senescent T cell accumulation at aged adipose tissue

Reference: Geriatr Gerontol Int . 2020 Feb;20(2):97-100. Cellular senescence and senescence-associated T cells as a potential therapeutic target. PMID: 31837250

Secondly, it is mentioned here that T cells, especially regulatory T cells (Treg) are particularly accumulated in the aged adipose tissue. However, Treg cells are immune suppressive T cell population that are supposed to inhibit inflammation instead of contributing to inflammation. Therefore, references supporting this viewpoint is needed to clarify this confusion, otherwise the Treg part should be deleted.

  1. Line 383 to 384 is about the increased M1 type Adipose tissue macrophages (ATMs) accumulation in the adipose tissue. This is an important source to age-related inflammation that contributing to the aged-related disease. Therefore, supporting references are supposed to be given in the text.

Author Response

This manuscript (MS) reviewed the current understanding of age-related change in lipid metabolism that may contribute to age-related diseases. It focused on the epigenetic changes and endocrine features of lipid metabolism during aging. Generally, it is ok for summarizing recent research program. However, there are several point concerning this article for its improvement:

Response to reviewers: Thanks for the detailed comments on the manuscripts. The reviewer’s suggestion was really helpful to improve the article.

1.As one of the ultimate purposes of the research of age-related lipid metabolism is to understand its influence in multiple age-related diseases, more of its links with diseases are supposed to be described. The following includes some reviews relating to this topic. These or other similar publications are recommended to be involved in the manuscript with brief descriptions on the mechanisms underlying age-related lipid metabolism and age-related diseases.

age-related lipid metabolism AND Alzheimer’s disease:

Reference-1: Int J Mol Sci. 2020 Feb; 21(4): 1505. Lipids and Alzheimer’s Disease. PMID: 32098382.

Refernce-2: Front Aging Neurosci. 2015 Oct 23; PMID: 26557086

age-related lipid metabolism AND Diabetes:

Reference: Lancet . 2010 Jun 26;375(9733):2267-77. Lipid-induced insulin resistance: unravelling the mechanism.PMID: 20609972

Response to reviewers: Thanks for comments. I added suggested reference in the manuscript.

2.Line 386 to 388 is about the T cells accumulation at aged adipose tissue contributing to regional and systemic inflammation. There are two points needed to be modified:

Firstly, references supporting this statement are supposed to be given. In aged immune system, there is an increased population of senescent T cells that contributing inflammation, therefore, the supporting reference here should include senescent T cell accumulation in aged adipose tissue. The following is one example reference about PD-1+ senescent T cell accumulation at aged adipose tissue

Reference: Geriatr Gerontol Int . 2020 Feb;20(2):97-100. Cellular senescence and senescence-associated T cells as a potential therapeutic target. PMID: 31837250

Secondly, it is mentioned here that T cells, especially regulatory T cells (Treg) are particularly accumulated in the aged adipose tissue. However, Treg cells are immune suppressive T cell population that are supposed to inhibit inflammation instead of contributing to inflammation. Therefore, references supporting this viewpoint is needed to clarify this confusion, otherwise the Treg part should be deleted.

Response to reviewers: Thanks for comments. I added suggested reference in the manuscript and modified some phrases to avoid confusion.

3.Line 383 to 384 is about the increased M1 type Adipose tissue macrophages (ATMs) accumulation in the adipose tissue. This is an important source to age-related inflammation that contributing to the aged-related disease. Therefore, supporting references are supposed to be given in the text.

Response to reviewers: Thanks for comments. I added suggested reference in the manuscript.

Reviewer 4 Report

This manuscript (ID: cells-1158549)aims to summarize the knowledge about the roles of lipids and the metabolisms in aging. The overall structure is concisely organized but covers comprehensive point of views (lipid accumulation, adipokines and lipid metabolites), thus, it is intriguing and will contribute to the research of aging.

There are only a few minor comments.

1. The figures are a little hard to follow. Figure legend would be appreciated especially for Figure 1, including the annotation of the items.

2. Although the manuscript referred to the role of lipid metabolism in skeletal muscle, it is not discussed about the role in cardiac muscle. It would be ideal if author could summarize the differences the role of lipids between in skeletal muscle and in heart. Alternately, adding a few citations to navigate to that topic will be appreciated.

3. Line 380:
“With age, ATMs accumulate in subcutaneous fat, but no significant change has been detected in visceral depots, suggesting that visceral adipose fat is a sentinel source of inflammatory cytokines. “
I didn’t get the logic why no change of ATMs in visceral depots provides this interpretation. It would be appreciated if author could add a few words to fill the logic gap.

4. Line 160:
Does “L- “of L-FABP stand for liver type?

Author Response

This manuscript (ID: cells-1158549)aims to summarize the knowledge about the roles of lipids and the metabolisms in aging. The overall structure is concisely organized but covers comprehensive point of views (lipid accumulation, adipokines and lipid metabolites), thus, it is intriguing and will contribute to the research of aging.

There are only a few minor comments.

  1. The figures are a little hard to follow. Figure legend would be appreciated especially for Figure 1, including the annotation of the items.

Response to reviewers: Thanks for the comment. I added figure legend for each figure to make it easy to follow.

  1. Although the manuscript referred to the role of lipid metabolism in skeletal muscle, it is not discussed about the role in cardiac muscle. It would be ideal if author could summarize the differences the role of lipids between in skeletal muscle and in heart. Alternately, adding a few citations to navigate to that topic will be appreciated.

Response to reviewers: Thanks for the comment. I added sentences that summarize the changes of lipid metabolism in heart aging.

  1. Line 380:

“With age, ATMs accumulate in subcutaneous fat, but no significant change has been detected in visceral depots, suggesting that visceral adipose fat is a sentinel source of inflammatory cytokines. “

I didn’t get the logic why no change of ATMs in visceral depots provides this interpretation. It would be appreciated if author could add a few words to fill the logic gap.

Response to reviewers: Thanks for the comment. I modified the expression.

  1. Line 160:

Does “L- “of L-FABP stand for liver type?

Response to reviewers: Thanks for the comment. I modified the expression.

Round 2

Reviewer 1 Report

The revised version of the manuscript entitled “Advances in understanding of the role of lipid metabolism in aging” by Ki Wung Chung looks sufficiently better as compared to the initially submitted version. Most of my comments and concerns have been addressed or explained. There are only little remarks that remain. They are as follows:

  • Section 2, line 106-107: the sentence “In addition to triglyceride metabolism, lipolysis in adipose tissue also changes during aging” is not good because lipolysis is lipid degradation and includes triglyceride metabolism and degradation too. It is better to remove this sentence.
  • The author uses both “ageing” and “aging” – see lines 103 and 104. This should be checked.
  • Section 2.2, line 229-231, the 2 sentences “The kidney has not classically been considered as an organ that mainly regulates whole-body metabolism. Indeed, the kidney does not actively participate in the metabolic interactions that occur among other major organs” are not good, because the kidney is involved in the regulation of whole-body homeostasis via regulating electrolyte balance, acid-base balance, and biosynthesis of hormones. Additionally, as the author describes further, metabolic processes occur in the kidney. These sentences can be removed.

Author Response

The revised version of the manuscript entitled “Advances in understanding of the role of lipid metabolism in aging” by Ki Wung Chung looks sufficiently better as compared to the initially submitted version. Most of my comments and concerns have been addressed or explained. There are only little remarks that remain. They are as follows:

Section 2, line 106-107: the sentence “In addition to triglyceride metabolism, lipolysis in adipose tissue also changes during aging” is not good because lipolysis is lipid degradation and includes triglyceride metabolism and degradation too. It is better to remove this sentence.

Response to reviewer: Thanks for your opinion. I made correction to avoid confusion.

The author uses both “ageing” and “aging” – see lines 103 and 104. This should be checked.

Response to reviewer: Thanks for your opinion. I changed ‘ageing’ to ‘aging’ .

Section 2.2, line 229-231, the 2 sentences “The kidney has not classically been considered as an organ that mainly regulates whole-body metabolism. Indeed, the kidney does not actively participate in the metabolic interactions that occur among other major organs” are not good, because the kidney is involved in the regulation of whole-body homeostasis via regulating electrolyte balance, acid-base balance, and biosynthesis of hormones. Additionally, as the author describes further, metabolic processes occur in the kidney. These sentences can be removed.

Response to reviewer: Thanks for your opinion. I removed the sentence you mentioned.